# Classification of Water Source in Coal Mine Based on PCA-GA-ET

**Zhenwei Yang** [1,2] , **Hang Lv** [1,2], **Xinyi Wang** [1,2,*], **Hengrui Yan** [1,2] **and Zhaofeng Xu** [1,2]

1   Institute of Resources & Environment, Henan Polytechnic University, Jiaozuo 454000, China; yangzhenwei2006@126.com (Z.Y.); lvhang0030@163.com (H.L.)
2   Collaborative Innovation Center of Coal Work Safety and Clean High Efficiency Utilization, Jiaozuo 454000, China
*   Correspondence: wangxy@hpu.edu.cn

**Abstract:** In recent years, inrush water has hampered the regular mining of coal mines, and the proper identification of the source of inrush water is critical to the prevention and management of water hazards in mines. This paper extracts the standard water chemistry discriminating ions $Na^++K^+$, $Ca^{2+}$, $Mg^{2+}$, $Cl^-$, $SO_4^{2-}$, and $HCO_3^-$ from observed water samples. An improved water source discrimination model is proposed which combines algorithms from data mining, classification models, and learning reinforcement. According to the Pearson correlation coefficient, $Na^++K^+$ has a strong correlation with $HCO_3^-$. To identify the major metrics, we performed principal component analysis (PCA), and the adaptive differential evolutionary genetic algorithm (GA) was utilized to optimize the depth of the extreme tree (ET) and the number of classifiers. Finally, the model distinguished 25 sets of studied samples from various water sources in the Pingdingshan coalfield. Comparative analysis demonstrated the efficacy of each stage of our work. PCA-GA-ET outperformed the conventional approaches, such as the support vector machine, BP artificial neural network, and random forest. The studies revealed that PCA-GA-ET can eliminate the information overlap between data and simplify the data structure and thereby improve the efficiency and accuracy of water source detection. We discovered that by utilizing the evolutionary algorithm to optimize parameters such as the depth of the extreme trees and the number of decision trees, we could get the model to converge faster and to be more stable and more accurate. The results suggest that PCA-GA-ET has good robustness and accuracy and can meet the needs of water source identification.

**Keywords:** water source discrimination; Pingdingshan coalfield; principal component analysis; genetic algorithm; extreme tree

## 1. Introduction

Due to the depletion of shallow coal resources, deep coal mining is frequently utilized in the coalfields of northern China [1,2]. Water damage in coal mines has become a significant problem that threatens miners' lives and property as coal mine depths have increased and coal mining work has accelerated. The management of mine water hazards typically begins at the source of water inrush, and determining the type of water inrush and the cause of water inrush correctly and efficiently is a critical stage in the prevention and control of water inrush in mines [3]. In recent studies, the methods used to determine the source of water inrush in mines have mainly included the water temperature and level method, laser-induced fluorescence method [4,5], and water chemical analysis method. One discriminative strategy is to develop discriminative functions for various aquifers based on the otherness of geochemical ion concentrations in aquifers combined with mathematical methods [6]. Distance discrimination, grayscale clustering discrimination, Fisher discrimination [7], and the fuzzy integrated assessment approach are examples of the commonly used mathematical methods. However, when the amount of data is significant enough, these approaches struggle to match the needs of water source discrimination.

Machine learning [8–10], a subfield of artificial intelligence, excels at classification and regression problems and involves research in statistics, probability theory, approximation networks, neural networks, and optimization theory. Water source discrimination can be seen as a classification problem [11–14]. Many scholars have achieved excellent results by incorporating machine learning classification methods, such as multiple logistic regression [15], Bayesian networks [16], the artificial neural network [17], the support vector machine [18], and so on, into the mine water discrimination problem, and the methods described above can solve the majority of the practical problems. Nevertheless, when the hydrological circumstances in the study area are complex and many types of water samples are intermingled, Bayesian networks cannot match the demand for numerous classifications. Artificial neural networks require a high number of samples to learn, and a lack of examples has a direct impact on the efficiency of BP artificial neural networks. The support vector machine's hyperparameters C (penalty coefficient) and gamma are difficult to determine. To avoid these issues, researchers chose appropriate algorithms to build multi-method fusion water source discernment models from the standpoints of data processing, model enhancement, and model evaluation. For example, Yan et al. established a water inrush identification model for marine metal mines using the adaptive differential evolutionary metropolis algorithm based on Bayesian theory with Markov chain Monte Carlo simulation as a parameter posterior distribution sampling calculation method, which provides guidance for the prevention and control of water inrush in subsea mining [19]. Wang et al. used the entropy weighting method (EWM) to weight chemical ions and combined it with hierarchical cluster analysis (HCA) to determine the source of water inrush [20]. The deep trust network (DBN) can identify implicit features in complicated hydrogeological information and efficiently filter missing and noisy data, with a prediction accuracy of 94% for water inrush sources, which is 24% higher than that of the typical BP artificial neural network [21]. Unfortunately, there are still shortcomings in the above studies: (1) the model construction is cumbersome and thus increases the complexity of the algorithm calculation; (2) the discriminant method does not take into account the information redundancy among water chemistry data, which makes the model training time longer; and (3) when small and medium-sized samples are input into the model learning, the model does not learn sufficiently, the training samples are less accurate, and the phenomenon of underfitting occurs.

To address the aforementioned issues, this study provides a new discriminative method (PCA-GA-ET) that removes the information overlap across data, has a fast training time, and is highly accurate. Raw data frequently have flaws that can impair the performance of machine learning models. Redundant features might waste computing resources and impair the model's generalization capabilities. The principal component analysis (PCA) approach is a popular dimensionality reduction technique that combines strongly linked data into fewer new features while eliminating the information overlap across features. In terms of model selection, the tree model can handle high-dimensional data and has great robustness, making it resistant to outliers and noise. Because the commonly used random forest technique causes 20% of the data to enter out-of-bag estimation during the training process and the small training sample results in insufficient model learning, we selected the extreme tree (ET) algorithm as the discriminant model. Extreme tree is a variant of random forest. It selects features at random for segmentation, which reduces the danger of overfitting. Furthermore, because of the uncertainty of feature selection, extreme trees can be taught faster than random forests. However, the performance of extreme trees is affected by factors such as the depth of the tree and the number of decision trees. The typical manual tuning strategy may overlook some parameter variations, causing the model to perform poorly. The shortcomings of extreme trees are overcome by applying genetic algorithms (GA) to determine the optimal solution of the extreme tree parameters in the search space. Genetic algorithms do not require information such as derivatives of the solution function and are thus appropriate for complex nonlinear situations. In this study, the PCA-GA-ET model provides the following advantages: (1) it has quick training time

and high recognition efficiency; (2) it can fit the data better when there are less data; (3) the PCA-GA-ET can effectively identify water sources and address the problem of complex hydrogeological circumstances.

## 2. The Theory of Methods

### 2.1. Principal Component Analysis

Principal component analysis transforms a set of potentially correlated variables into a new set of linearly uncorrelated variables by means of orthogonal transformation. The new variables obtained by the transformation are called principal components. They are able to keep the original information that is to be revealed unchanged in terms of the information that is expressed. The main goal of principal component analysis is to identify hidden patterns in the data and reduce the noise and redundancy in the data, eliminate information overlap between variables, and simplify the data structure. The steps of PCA are calculated as follows:

Let $X_1, X_2, \ldots, X_p$ be the variables of the whole sample and make a linear transformation $Y = AX$, that is:

$$\begin{cases} Y_1 = a_{11}X_1 + a_{12}X_2 + \ldots a_{1p}X_p, \\ Y_2 = a_{21}X_1 + a_{22}X_2 + \ldots a_{2p}X_p, \\ \qquad\qquad \ldots \\ Y_p = a_{p1}X_1 + a_{p2}X_2 + \ldots a_{pp}X_p, \end{cases} \tag{1}$$

Among them, $Y_1, Y_2, \ldots, Y_P$ represents the new main component, $X_1, X_2, \ldots, X_P$ represents the original characteristics, and $a_{i1}, a_{i2}, \ldots, a_{ip}$ represents the linear combination coefficient.

In addition, the following conditions are met: $a_{i1} + a_{i2} + \ldots + a_{ip} = 1 (i = 1, 2, \ldots, p)$; $Y_1$ has the largest variance in the linear combination; the $Y_1, Y_2, \ldots, Y_P$ variance decreases in turn; $Y_i (i = 1, 2, \ldots, p)$ is not correlated with the others. Calculate the variance contribution of each principal component and rank it according to the variance contribution and retain the components that can overview the entire sample information (the number of principal components is less than $P$), so as to reduce the quantity of data.

### 2.2. Genetic Algorithm

Genetic algorithms (GA) are a method of searching for optimal solutions by simulating the natural evolutionary process, mainly using probabilistic search methods, which operate on structural objects and automatically acquire and guide the search space for optimization. Starting from an initialized population of possible feasible solutions to the problem, iterative "replication", "crossover", and "mutation" operations are performed among the feasible solutions. The adaptation value of each feasible solution is calculated, and the optimal individual in each evolving population is obtained according to the evolutionary principle of "survival of the fittest". The algorithm is implemented as follows:

(1) In order for the algorithm to perform a fast solution search, it is necessary to map the space where the feasible solutions are located to the genotype space. The decoding and encoding form of the feasible solution is generally characterized by a binary string. The length of the binary string encoding symbol is related to the solution accuracy sought by the problem. Let a parameter take a range of values $[U_{\min}, U_{\max}]$, an interval length $L = U_{\max} - U_{\min}$, and a string length $l$. The relationship when encoding the parameter is as follows:

$$00000000\cdots00000000 = 0, \quad 00000000\cdots00000001 = 1, \quad \ldots \quad 11111111\cdots11111111 = 2^l - 1$$
$$\qquad\quad \downarrow \qquad\qquad\qquad\qquad\quad \downarrow \qquad\qquad \ldots \qquad\qquad\qquad \downarrow$$
$$\qquad\quad U_{\min} \qquad\qquad\qquad\qquad U_{\min} + \delta \qquad\quad \ldots \qquad\qquad\qquad U_{\max}$$

Then, the encoding precision of binary encoding is:

$$\delta = \frac{L}{2^l - 1} \tag{2}$$

Among them, $\delta$ is the coding accuracy. Let a parameter take a range of values $[U_{\min}, U_{\max}]$, an interval length $L = U_{\max} - U_{\min}$, and a string length $l$.

Assume that the individual solution $X : b_l b_{l-1} b_{l-2} \ldots b_2 b_1$ is decoded as:

$$x = U_{\min} + \left(\sum_{i=1}^{l} b_i \cdot 2^i - 1\right) \cdot \delta \tag{3}$$

Then, $x$ is the optimal solution obtained, $U_{\min}$ is the minimum solution that can be obtained by the parameter, and $b_i$ is the binary number of the optimal solution.

(2) Generate the initial population, set the maximum evolutionary algebra $T$, population size $M$, crossover probability $P_c$, and mutation probability $P_m$, and randomly generate $M$ individuals as the initial population $P_0$.

(3) The fitness function is defined to evaluate the advantages and disadvantages of the feasible solution. Different problems have different definitions of the fitness function. The fitness function Micro-F1 is introduced, and its calculation formula is as follows:

Firstly, calculating the recall rate:

$$Recall_m = \frac{\sum_{i=0}^{n} TP_i}{\sum_{i=0}^{n} TP_i + \sum_{i=0}^{n} FN_i} \tag{4}$$

Secondly, calculating the precision:

$$Precision_m = \frac{\sum_{i=0}^{n} TP_i}{\sum_{i=0}^{n} TP_i + \sum_{i=0}^{n} FP_i} \tag{5}$$

Among them, $TP_i$ is the true positive of category $i$, $FN_i$ is the false negative of category $i$, and $FP_i$ is the false positive of category $i$.

Finally, substitute the calculated Recall and Precision into the following formula to get Micro-F1:

$$Micro - F1 = 2\frac{Recall_m \times Precision_m}{Recall_m + Precision_m} \tag{6}$$

(4) Regenerating individuals are chosen based on fitness, and individuals with low fitness are discarded. Selection, crossover, and variation generate new populations, and the elite retention strategy converges globally to the ideal solution and outputs the optimal solution hyperparameter values, as shown in Figure 1.

### 2.3. Extreme Tree (ET)

The extreme tree (ET) algorithm is similar to the random forest (RF) technique in that both use multiple binary trees. The distinction is that RF uses a bagging model to pick characteristics. ET employs the whole sample set, and the features are chosen at random, enhancing the randomness of the model and making it, to some extent, better than RF. ET can be represented by {T (K, X, D)}, where T represents the final classifier model, and sample X= {$x_1, x_2, \ldots, x_N$} trains multiple binary tree models and votes between binary trees to determine the classification category. The specific steps are as follows:

(1) Given the original dataset D, the number of samples N, and the number of features M, each binary tree uses all the samples for training.

(2) The CART algorithm generates a binary tree by calculating the Gini coefficient (Gini) of the splitting features. When the binary tree nodes are split, randomly select m features from the M features in order to split and find the appropriate attribute split for each node. After splitting, the nodes repeat (2) to produce a binary tree.

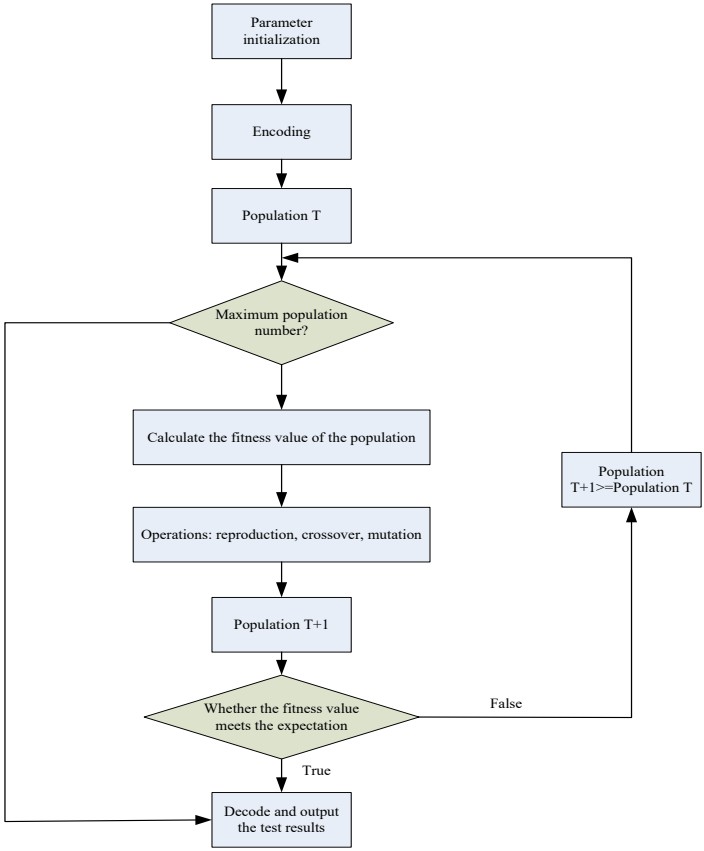

**Figure 1.** Steps for GA to obtain the optimal solution.

(3) Repeat (1) and (2) *K* times and finally generate K binary trees to form an extreme random tree.

(4) Input the validation samples into the ET that was generated. To obtain the identification results, count all the binomial trees. Using the voting selection process, generate the final prediction results.

The tree depth (DP) and the number of trees (ES) are known to be hyperparameters of the ET model from the creation process. Second, changing the minimum number of samples on the split nodes and the minimum number of samples on the leaf nodes influences the model's accuracy. As a result, GA is primarily responsible for optimizing the aforementioned settings.

## 3. Data Collection and Preprocessing

### 3.1. Study Area

The Pingdingshan coalfield is situated in the middle section of Henan Province in northern China, around 113°00–114° E and 33°30′–34°00′ S. The Pingdingshan coalfield's coal-bearing layers are Late Paleozoic Permian coal-bearing lithologies. The majority of the coal-bearing sediments are Permian sediments composed of sandstone, siltstone, and carbonaceous shale. Neoproterozoic, Paleoproterozoic, and Quaternary deposits cover the Pingdingshan coalfield. The entire sequence is underlain by Cambrian karst limestone, which geologically creates a succession of complicated folded tectonic formations running north to west. It is accompanied by tensor-torsional and compression-torsional fractures oriented north to west, as well as tensor-torsional fractures oriented north to east. The annual rainfall ranges from 373.9 to 1323.6 mm, with an average of 742.6 mm, and the rainy season is primarily focused on July, August, and September. The main threatening water-filled aquifers in the study area from top to bottom include Quaternary pore aquifers with a unit surge of 0.00017 L/s m and a permeability coefficient of 0.000626 m/day; Carboniferous tuff aquifers with a unit surge of 0.00018–0.3569 L/s m and a permeability coefficient of

0.0076–3.047 m/day; and Cambrian tuff aquifers with a unit surge of 2.27–26.62 L/s m and a permeability coefficient of 1.092–7.42 m/day. Mine extraction is directly influenced by surface water and Permian sandstone aquifers. It is critical to investigate the differences in the hydrochemical properties of diverse water-filled water sources and to develop matching identification models for judging water sources in a timely manner. Figure 2 depicts the geological structure of the Pingdingshan coalfield:

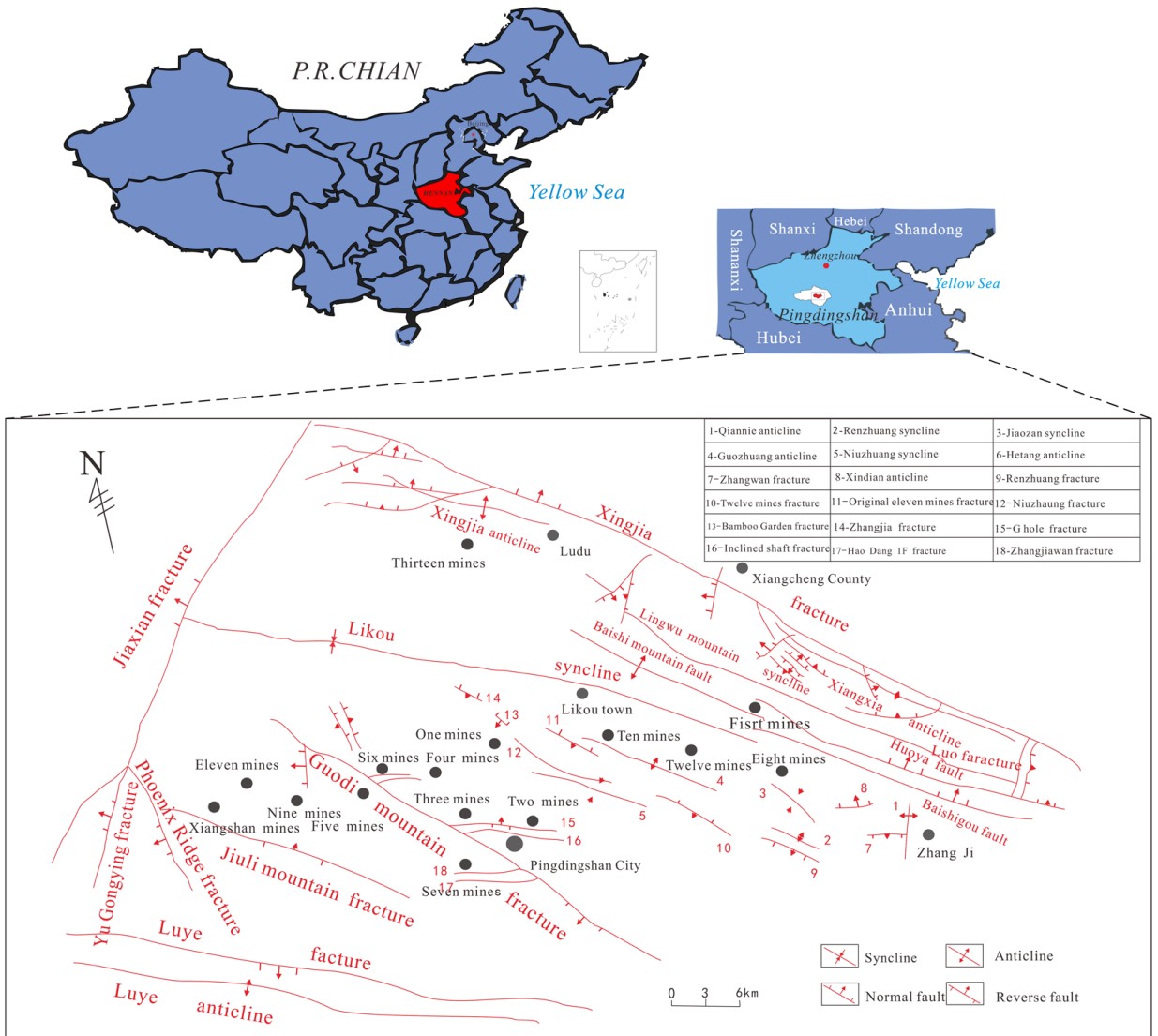

**Figure 2.** Geological structure map of Pingdingshan coalfield.

### 3.2. Statistical Analysis and Data Processing

From February 2017 to December 2021, 124 groups of mine water source data were collected in the Pingdingshan coalfield. These include 19 groups of surface water data (I); 16 groups of Quaternary pore water data (II); 44 groups of Carboniferous tuff karst water data (III); 22 groups of Permian sandstone water data (IV); and 23 groups of Cambrian tuff karst water data (IV). In order to conform to computer language processing, they were simplified to [0, 1, 2, 3, 4] after unique thermal coding. The water sources were sent to a qualified water quality testing facility, and the general hydrogeochemical variables $Na^+ + K^+$, $Ca^{2+}$, $Mg^{2+}$, $Cl^-$, $SO_4^{2-}$, and $HCO_3^-$ were extracted from the sample water samples. These are hereafter referred to as $X1$, $X2$, $X3$, $X4$, $X5$, and $X6$. The statistics data are shown in Table 1.

**Table 1.** Categories of water source and their One-Hot encoding.

| Category | Sample Capacity | Target | One-Hot Encoding |
|---|---|---|---|
| Surface water | 19 | 0 | [1.0.0.0.0] |
| Quaternary pore water | 16 | 1 | [0.1.0.0.0] |
| Carboniferous limestone karst water | 44 | 2 | [0.0.1.0.0] |
| Permian sandstone water | 22 | 3 | [0.0.0.1.0] |
| Cambrian limestone karst water | 23 | 4 | [0.0.0.0.1] |

The Piper diagram in Figure 3 can be used to examine the link between the five different types of water sources and their ionic properties. The Piper diagram simply depicts the general hydrochemical properties of each type of water sample. The distribution of water samples in the diagram has several intersection sites. The old method does not effectively identify the type of water samples; so, the water chemical ion properties must be used as input variables to train the water source discriminating model.

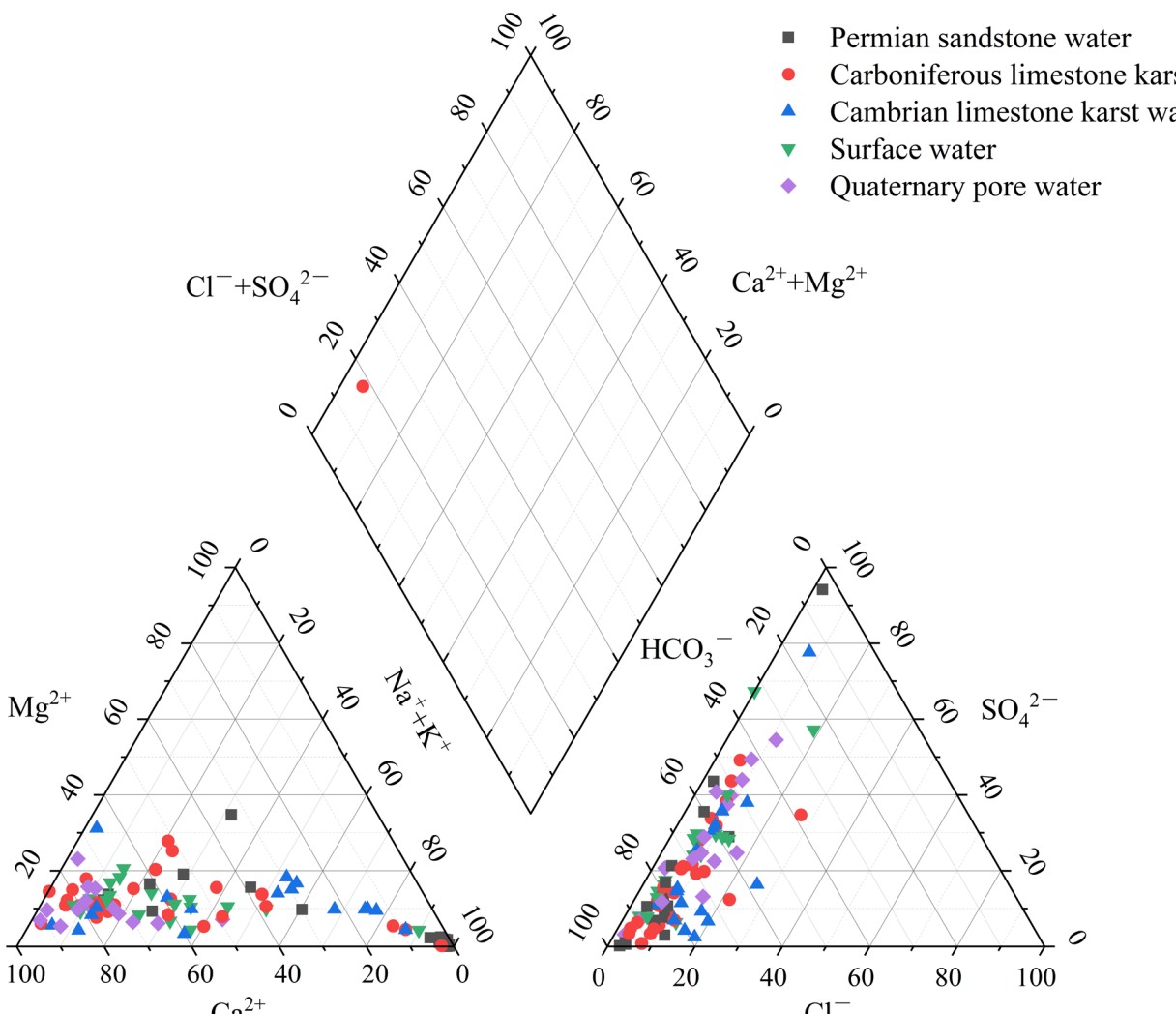

**Figure 3.** Piper diagram of hydrochemistry of Pingdingshan coal water sample.

Figure 4 depicts the distribution of water chemical concentrations, and the distribution of $X1$ varies from 0.14 to 1109.74 (mg·L$^{-1}$). $X1$ was more densely spread in the range of 0.14 to 100.00 (mg·L$^{-1}$). $X2$ ranged from 2.4 to 417.33 (mg·L$^{-1}$), and there was a tendency to show a double-wave peak. The $X3$ distribution ranged from 0 to 173.4 (mg·L$^{-1}$). $X4$ had a similar distribution to $X3$, but the $X4$ waveform shifted to the left and had a right-skewed

distribution with a larger peak than that of *X3*. The distribution of *X5* was comparable to that of *X1*. The *X6* distribution varied from 52.48 to 2498.77 and was much greater than the other ion concentration distributions. This tends to increase the complexity of the calculation by a substantial difference in magnitude and can influence the discriminative model's accuracy. As a result, the sample data must be standardized so that the data are proportionally constrained to the range [0, 1], thereby reducing the harmful impacts generated by anomalous data. The following is the standardizing equation:

$$X^* = \frac{X - Min(X)}{Max(X) - Min(X)} \tag{7}$$

where $X^*$ is the standardized value of the variables; *Max* is the maximum value of the sample data; and *Min* is the minimum value of the sample data.

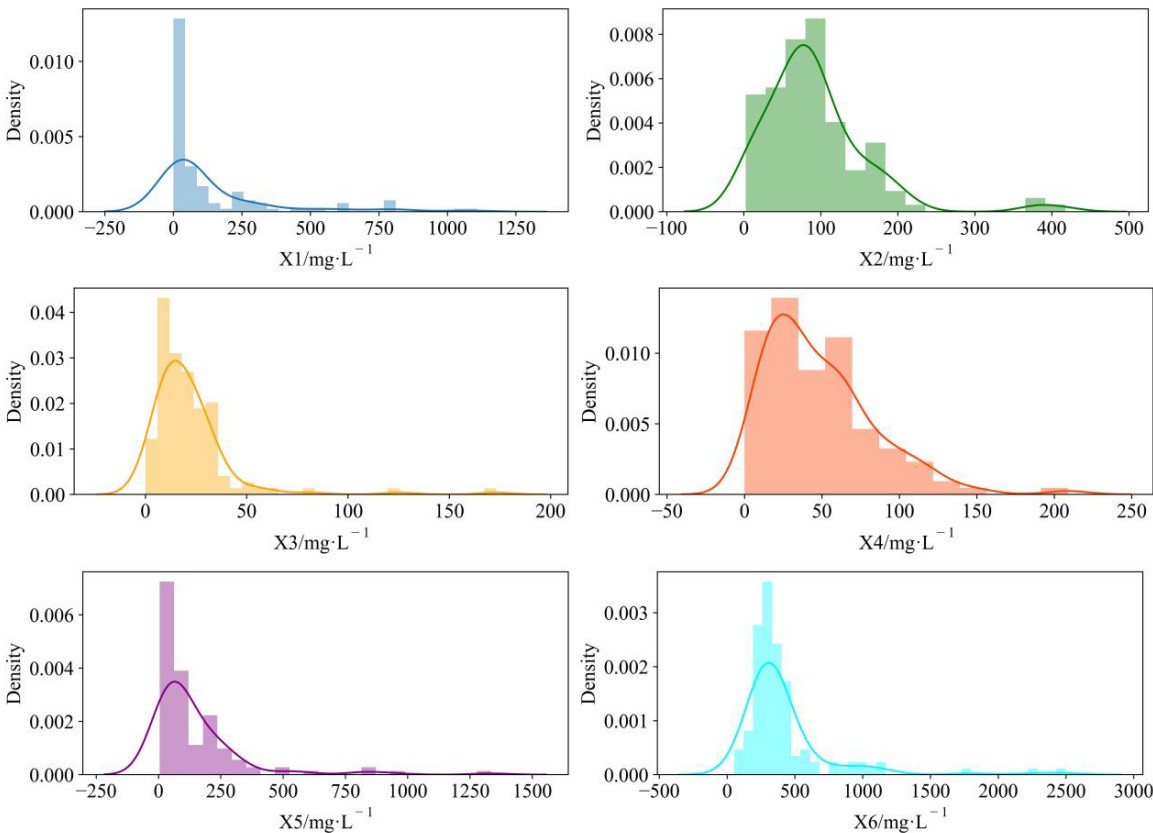

**Figure 4.** Kernel density distribution of the hydrochemical variables *X1*, *X2*, *X3*, *X4*, *X5*, and *X6*.

The Pearson correlation coefficient can be used to determine the degree of correlation between the six chemical components of water. Table 2 displays the correlation coefficients for the water chemistry components. According to Table 2, the correlation coefficient between *X1* and *X6* is 0.8849. These two variables have a clear relationship. To avoid overlapping information amongst the indicators, which causes the model to misunderstand water sources, PCA was used to reduce the dimensionality of the input sample data.

**Table 2.** Statistics of water source types in Pingdingshan.

|  | **X1** | **X2** | **X3** | **X4** | **X5** | **X6** |
|---|---|---|---|---|---|---|
| X1 | 1.0000 |  |  |  |  |  |
| X2 | −0.3316 | 1.0000 |  |  |  |  |
| X3 | 0.0129 | 0.4928 | 1.0000 |  |  |  |
| X4 | 0.4187 | 0.1846 | 0.3257 | 1.0000 |  |  |
| X5 | 0.2354 | 0.3780 | 0.5666 | 0.1947 | 1.0000 |  |
| X6 | 0.8849 | −0.3039 | −0.1197 | 0.4391 | 0.0206 | 1.0000 |

The variance contributions of the five principal components were obtained after the PCA dimensionality reduction. The first two variance contributions were 0.53772683 and 0.3507404, and the last three principal components were 0.10706789, with a cumulative variance contribution of 99.55%. The PCA mathematical model is as follows:

$$Y_1 = -0.04864666X_1 + 0.16447242X_2 + 0.04444455X_3 + 0.04098346X_4 + 0.82772985X_5 - 0.53083957X_6 \quad (8)$$

$$Y_2 = 0.78442504X_1 - 0.59249409X_2 - 0.05155806X_3 + 0.00705547X_4 + 0.0603243X_5 - 0.16516984X_6 \quad (9)$$

$$Y_3 = 0.33084572X_1 + 0.54580671X_2 + 0.07496453X_3 + 0.19127722X_4 - 0.47042781X_5 - 0.57369605X_6 \quad (10)$$

$$Y_4 = 0.11204105X_1 + 0.07087432X_2 + 0.13592432X_3 + 0.91228671X_4 + 0.15150198X_5 + 0.32973977X_6 \quad (11)$$

$$Y_5 = -0.50910722X_1 - 0.56199211X_2 + 0.05928003X_3 + 0.32146666X_4 - 0.25875703X_5 - 0.50116321X_6 \quad (12)$$

where $Y_1$, $Y_2$, $Y_3$, $Y_4$, are $Y_5$ are freshly created features, and $X_1$, $X_2$, $X_3$, $X_4$, $X_5$, and $X_6$ are original features.

## 4. Establishment of Water Source Discrimination Model

This experiment was carried out programmatically on Spyder software. The ET model was built using the Sklearn machine-learning platform. The GA algorithm was built using the Geatpy genetic algorithm library. The sample data following PCA processing were cut by 8:2, with 80% of the data going into model learning and 20% going into model testing. The learning samples includes 16 groups of surface water, 14 groups of Quaternary pore water, 35 groups of Carboniferous chert karst water, 18 groups of Permian sandstone water, and 16 groups of Cambrian chert karst water. The specific model implementation phases are as follows:

Step1: The maximum evolutionary generation MaxGen = 50, the population size N = 50, the differential evolution parameter F = 0.5, the crossover probability XOVR = 0.8, the maximum upper limit of evolution MTC = 20, the fitness function Micro-F1 mentioned in the first section, and the dimensionality of the solution D = 4 are among the initialized GA population parameters.

Step2: The parameters of GA optimization ET comprise the number of estimators (ES), binomial tree depth (DP), the minimum number of samples of leaf nodes (MSL), and the minimum number of samples of split nodes (MSS), which are set in the ranges of 10~200, 10~30, 2~5, and 2~5, respectively.

Step3: Based on the adaptation value, the GA determines the best solution. If the GA does not identify the best answer this time, the initial population is forced to "crossover" and "mutate" to create a new population. If the model's optimal solution is found, the GA decodes it and returns the hyperparameter values.

During the optimization process, the GA repeatedly jumps out of the local optimal solution. The average target value rises iteratively with population turnover until it reaches its maximum in the 12th generation, after which the target value remains constant (Figure 5).

The ET parameters that yield the highest target value are ES = 41, DP = 14, MSL = 2, and MSS = 3. When the ET optimum parameters are used, the training sample's accuracy is 99%.

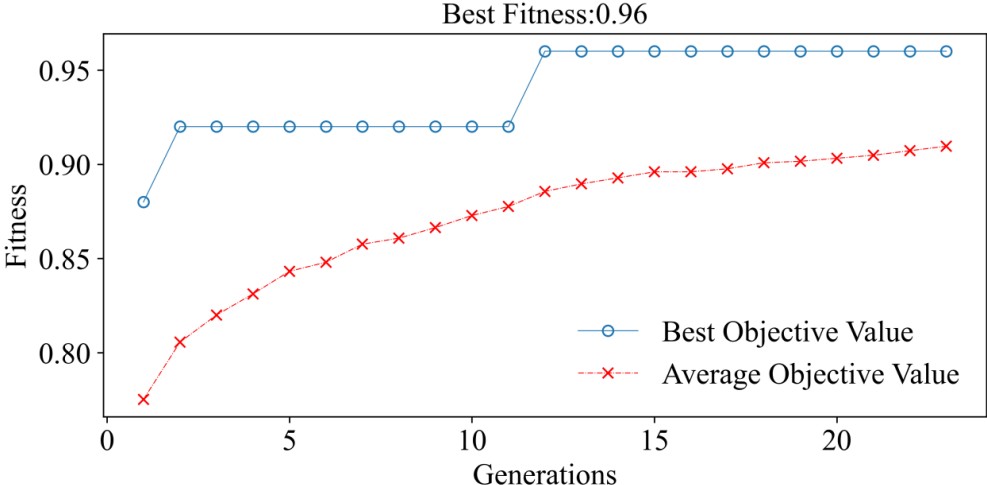

**Figure 5.** GA optimized ET fitness curve.

*Comparison of Models*

To further validate the PCA-GA-ET model's performance and correctness, the ET, PCA-ET, GS-RF, MLP, and PSO-SVM models used identical test samples. The model correctness and root mean square error (RMSE) were employed as discriminant criteria for comparing model outputs. The initialized particle swarm algorithm (PSO) parameters were inertia factor W = 0.5; learning factor C1 = 0.2, C2 = 0.5; number of iterations = 20; and population size N = 100. The SVM hyperparameters were optimized for PSO. Grid search (GS) optimizes the RF settings. Table 3 lists the specific parameters:

**Table 3.** MLP, PSO-SVM, and GS-RF comparison model parameters (The maximum number of features of the RF model was simplified to MF, and the number of hidden layers, neurons per layer, and learning rate of the MLP model were simplified to HL, NL, and LR).

| GS-RF | | | | | |
|---|---|---|---|---|---|
| Parameter | Algorithm | MF | ES | DP | MSL | MSS |
| Value | CART | 6 | 30 | 16 | 2 | 1 |
| **PSO-SVM** | | | | | |
| Parameter | Kernel | C | Gamma | Cache_size | Class_weight | Tol |
| Value | RBF | 6 | 8 | 200 | 1 | 0.001 |
| **MLP** | | | | | |
| Parameter | Activation | HL | NL | LR | Batch_size | Alpha |
| Value | Relu | 2 | 15 | 0.001 | 64 | 0.0001 |

The same test samples were fed into ET, PCA-ET, GS-RF, MLP, PSO-SVM, and PCA-GA-ET to compare their performance. (Figure 6). In addition, the learning samples validated the model's empirical error. (Figure 7). The model's evaluation indicators were RMSE and accuracy, and the RMSE is calculated as follows:

$$RMSE = \sqrt{\frac{1}{N}\sum_{i=1}^{N}(\hat{y}_i - y_i)^2} \tag{13}$$

where N is the number of observations, $\hat{y}_i$ is the predicted value, and $y_i$ is the true value.

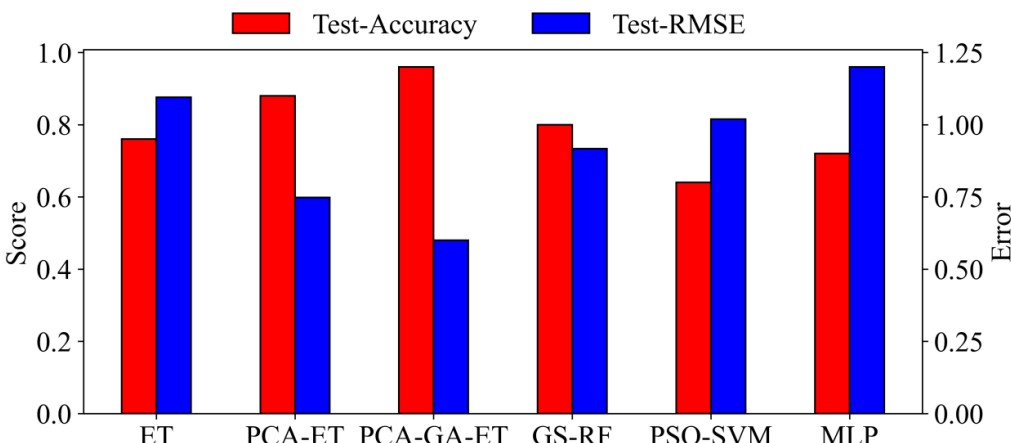

**Figure 6.** Performance comparison of ET, PCA-ET, PCA-GA-ET, GS-RF, PSO-SVM, and MLP in test samples.

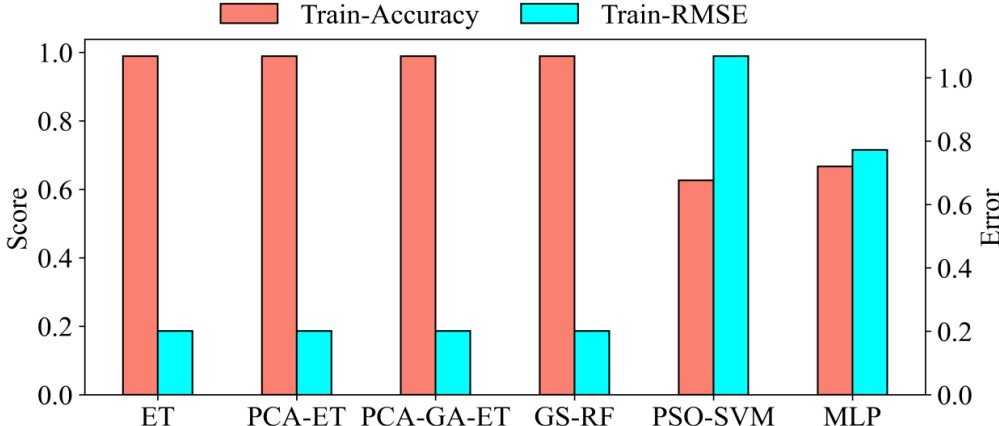

**Figure 7.** Performance comparison of ET, PCA-ET, PCA-GA-ET, GS-RF, PSO-SVM, and MLP in learning samples.

Figure 6 clearly compares the performance of each model for the examined samples. The ET accuracy was only 0.76, making it difficult to meet the needs of water source differentiation. PCA dimensionality reduction decreased model information redundancy and improved model accuracy, but it still did not approach optimal performance. The accuracy of the GA-optimized PCA-ET model increased considerably. The PCA-GA-ET model's accuracy was 0.96, and the RMSE was only 0.6. The GS-RF generalization error was 0.917, the PSO-SVM RMSE was 1.019, and the MLP error was 1.2. When compared to other models, the author's PCA-GA-ET model had the lowest generalization error, proving the model's dependability and validity.

The accuracy of the learning samples after they were entered into the model demonstrates that the tree model better matched the learning samples (Figure 7). The RMSE of the training samples in PSO-SVM was substantially bigger than that of the test samples, indicating that the model PSO-SVM was overfitting. In MLP, on the other hand, the error of the test sample was greater than that of the training sample, and the model was underfitting. The inter-model comparison reveals that PCA-GA-ET was more resistant to overfitting.

## 5. PCA-GA-ET Model Performance Verification

From the water inrush source samples of Pingdingshan coalfield, 25 groups of samples that did not enter the training model learning were selected for model validation. Among them, the 25 groups of verification samples included 3 groups of surface water, 2 groups of Quaternary pore water, 9 groups of Carboniferous limestone karst water, 4 groups of Permian sandstone water, and 7 groups of Cambrian limestone karst water.

Figure 8 depicts the discrimination outcomes. In the figure, 24 groups of samples were accurately classified, while just one group of Quaternary pore water was mistakenly predicted as Cambrian limestone karst water, for a total accuracy of 96%. The PCA-GA-ET identification model was effective. The identification inaccuracy could be because coal mining is less threatened by water inrush from the Quaternary aquifer. Because the quantity of Quaternary aquifer water source samples was restricted, the model did not learn enough about it. Raising the number of samples may help to eliminate the incorrect identification of Quaternary pore water. The preceding demonstrates that the water source identification model based on PCA-GA-ET is more accurate and stable and that it can match the requirement for water source identification.

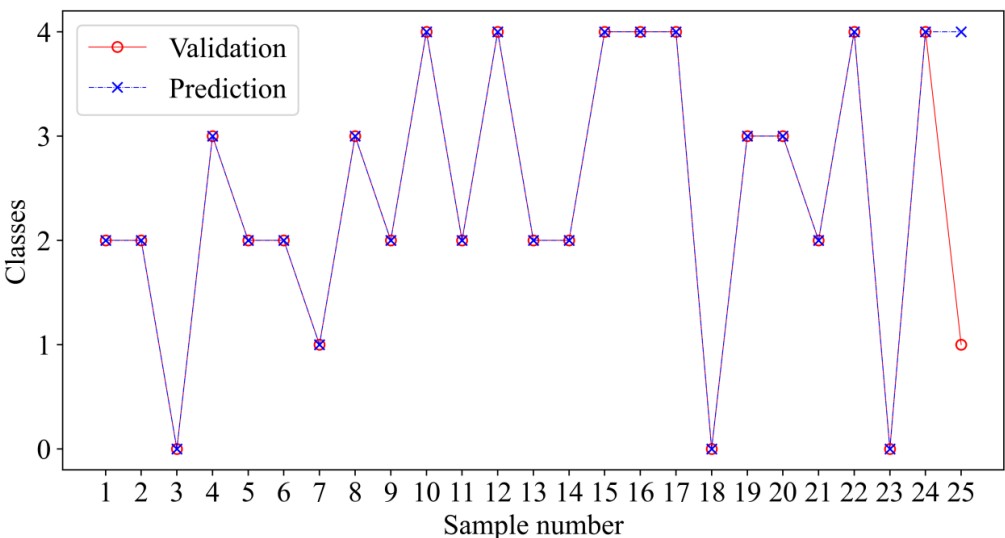

**Figure 8.** The discriminant results of the PCA-GA-ET water source identification model on the validation samples.

The CART algorithm can compute the size of a variable's Gini coefficient to rank the variables, representing the discriminative ability of different indicators, i.e., the larger a variable's Gini coefficient, the greater the relevance. As shown in Figure 9, $X2 > X3 > X1 > X4 > X5 > X6$. After calculating the Gini coefficient, $\text{VIM}_{Ca^{2+}}^{(Gini)} = 0.26859513$, $\text{VIM}_{Mg^{2+}}^{(Gini)} = 0.18861467$, $\text{VIM}_{Na^{+}+K^{+}}^{(Gini)} = 0.17606162$, $\text{VIM}_{Cl^{-}}^{(Gini)} = 0.15847374$, $\text{VIM}_{SO_4^{2-}}^{(Gini)} = 0.11474677$, $\text{VIM}_{HCO_3^{-}}^{(Gini)} = 0.09350808$. It can be seen that $Ca^{2+}$, $Mg^{2+}$, and $Na^{+}+K^{+}$ have strong discrimination abilities.

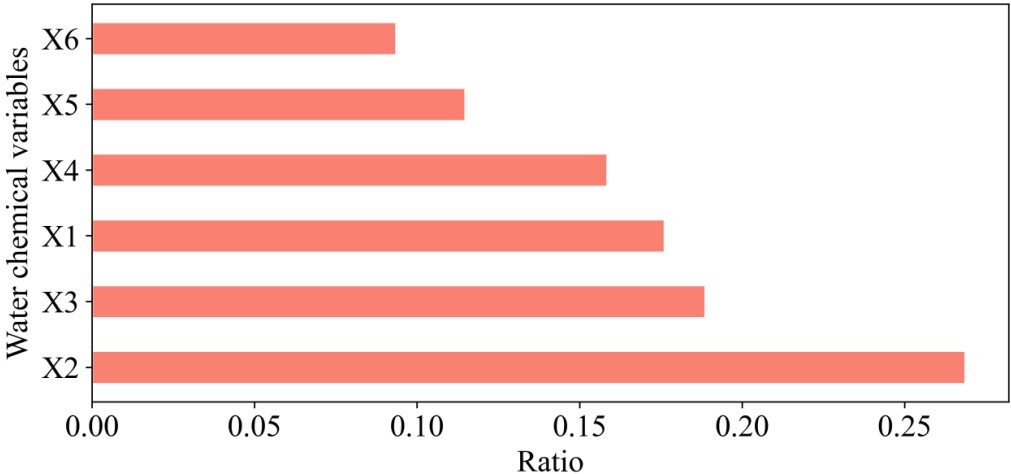

**Figure 9.** Comparison of variable significance calculation results.

## 6. Conclusions and Outlook

This work used the Pingdingshan coalfield as the research object; it integrated principal component analysis, extreme tree, and genetic algorithms to differentiate distinct water source samples and then presented the PCA-GA-ET water source discrimination method. The primary goal was to develop an intelligent mine water identification technology to prevent inrush water hazards in coal mines. The following are the primary conclusions:

(1) The kernel density plots of the coalfield water samples show that the density distributions of the water chemical data vary. The standardized data processing removes the effect of data magnitude on the model. Furthermore, when there are more data, the Piper plot is ineffective at distinguishing different water sources. Principal component analysis can help improve the quality of data. PCA-ET is more accurate than ET, demonstrating that PCA removes information redundancy in the sample data and enhances model correctness.

(2) In the Sklearn framework, a machine learning model for water source discrimination in mines under difficult hydrogeological settings is constructed. The parameters such as the number of estimators (ES), binary tree depth (DP), the minimum number of samples of leaf nodes (MSL), and the minimum number of samples of split nodes (MSS) are altered during the process of splitting the binary tree of the model. We set the model's parameters to improve the model's classification effect.

(3) The genetic algorithm (GA) optimizes the PCA-ET model's hyperparameters and secondary parameters. The model's best viable solution is obtained when the parameters are ES = 41, DP = 14, MSL = 2, and MSS = 3. The comparison of the PCA-ET and PCA-GA-ET models validates the GA algorithm's applicability to the ET model and increases the model's performance. The PCA-GA-ET model developed in this work is superior to the tree model RF. When the PCA-GA-ET model is compared to the regularly used SVM and MLP models, the PCA-GA-ET model fits the data better, proving the PCA-GA-ET model's reliability.

(4) As validation samples, we chose 25 groups of samples that did not participate in model training. With a model accuracy of 96%, 24 groups of samples were properly predicted, and only one group was mistakenly labeled as Cambrian water. The source of the water inrush could be identified using the PCA-GA-ET model. We assessed the Gini coefficient for each chemical feature, and $Ca^{2+}$ had the highest, indicating that $Ca^{2+}$ had the best discrimination capacity in identifying water inrush sources.

The research results reveal that the created machine learning model has excellent accuracy and application in mine water source discrimination, which gives guidance for mine water source discrimination. The structure of the original data was simplified using principal component analysis, which eliminated data interference with the model. We tried many machine learning models during the model selection process and discovered that tree models fit the existing data better. Random forest is a popular tree model; however, it has several limitations. Random forest, for example, has a lengthy training period, is prone to overfitting, and is sensitive to noisy data. ET divides the nodes using more random values, which reduces the variance of the model and so overcomes the shortcomings of random forests. The model's performance is heavily influenced by characteristics such as tree depth and the number of classifiers. We employed a genetic algorithm to obtain the best value for the parameters faster. The improved accuracy and stability of PCA-GA-ET made it adaptable to a variety of mining regions with similar hydrogeological characteristics. This method of identification is speedier and more convenient for researchers. The main contribution of this study lies in the determination of groundwater sources in hydrogeological complex mines using the PCA-GA-ET technique, overcoming the shortcomings of ET in the water source identification problem and providing a new way of thinking for mine water identification methods using machine learning multi-method fusion. Despite the fact that the source of mine water was established, there are still some issues. First, this technology has not been tried in mines with varying geological conditions. Furthermore, if the chemical concentration of the groundwater is not present in the concentration distribution of the study sample, the approach is invalid. The following research should attempt to apply

machine learning methods to different mining sites with different geological conditions, as well as use GIS methods to understand the geological conditions surrounding the water samples and to closely integrate the hydrological conditions of groundwater. Furthermore, the sample database should be built up by collecting as many samples as feasible. The model learned enough from the sample database to increase generalization and, to a large extent, to avoid mine water breakout mishaps. Finally, it is a worthwhile research approach to investigate an accurate, rapid, and stable intelligent system for identifying mine water sources.

**Author Contributions:** H.L. and H.Y. designed the model; Z.Y. and X.W. collected the mine water samples; Z.X. performed model training and data analysis; H.L. and Z.Y. wrote the manuscript. All authors have read and agreed to the published version of the manuscript.

**Funding:** This research was funded by State Key Laboratory of Development and Comprehensive Utilization of Coking Coal Resources grant number 41040220201308, National Natural Science Foundation of China grant number 41972254. And The APC was funded by Natural Science Foundation of Henan Province, the China Postdoctoral Science Foundation, Key Scientific Research Projects of Higher Education Institutions of Henan Province, and Fundamental Research Funds for the Universities of Henan Province.

**Data Availability Statement:** The data presented in this study are available on request from the corresponding author. The data are not publicly available due to privacy.

**Acknowledgments:** The authors acknowledge the reviewers' time and efforts in reviewing the article. Their positive feedback and encouragement have been instrumental in enhancing the quality of the final manuscript. The authors also thank for the hard working of journal editors.

**Conflicts of Interest:** The authors declare no conflict of interest.

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
