# Peer review of "Classification of Water Source in Coal Mine Based on PCA-GA-ET"

_water, doi:10.3390/w15101945_

Round 1

Reviewer 1 Report

The paper presents the use of PCA-GA-ET for classification of mine water based on the different ions as input features. The research results have some guiding significance to the engineering practice. Overall, this is an excellent report on very thorough research, and I approve of its publication if the followings details of my comments on the manuscript are improved carefully.

1. The title needs to be reformulated. It is better to delete word “inrush”, since there is not inrush water sample used in the study.

2. The initials of the Keywords should be small letter.

3. The number of the sections should be renumbered. Since the first section ‘Introduction’ has not been numbered, the following sections are numbered from number 1 to 5.

4. The second paragraph of the Introduction section is too long to be understood, and I would suggest that divide it into two paragraphs: one is for the research status, another is for virtue of the machine learning method.

5. In the last paragraph of the Introduction, I would suggest to first list the contributions rather than mentioning within the text and separate out the organization at the end.

6. Authors have introduced the potential limitations of existing methods, however, they do not illustrated that how AI can come-in to facilitate those limitations.

7. Methods such as PCA, GA and ET has been used to classifying water sources. What is the reason for using the three methods? How these methods are coupled in the study?

8. Regarding the datasets, it would be nice to see the correlation map between the input features and target classes.

9. The format of the manuscript should be improved carefully.

10. There is too much content in the last paragraph, and it should be written concisely.

11. What are the limitations of the presented research? And what the implications of the proposed approach? Authors may think of providing a practical use-case demonstrating how this trained model may be used in practice.

Author Response

Response to Reviewer 1 Comments

Re: Manuscript ID:water-2351772 and Title:Classification of water source in coal mine based on PCA-GA-ET

Response to Reviewers

Dear reviewer 1:

Thank you for your letter and the reviewer’s comments concerning our manuscript entitled “water-2351772” . Those comments are valuable and very helpful. We have read through comments carefully and have made corrections. Based on the instructions provided in your letter, we uploaded the file of the revised manuscript. Revisions in the text are shown using red highlight for additions, and strikethrough font for deletions. Those changes are highlighted within the manuscript. Please see below, in blue, for a point-by-point response to the reviewer’s comments and concerns. All page numbers refer to the revised manuscript file with tracked changes.

Reviewer's Comments to the Authors:

The paper presents the use of PCA-GA-ET for classification of mine water based on the different ions as input features. The research results have some guiding significance to the engineering practice. Overall, this is an excellent report on very thorough research, and I approve of its publication if the followings details of my comments on the manuscript are improved carefully.

  • The title needs to be reformulated. It is better to delete word “inrush”, since there is not inrush water sample used in the study.

Author response: Thank you for pointing this out. I deleted the word 'inrush' from the title on page 1.

  • The initials of the Keywords should be small letter.

Author response: Thank you for pointing this out. I changed the first letter of keywords to lower case on page 1.

  • The number of the sections should be renumbered. Since the first section ‘Introduction’ has not been numbered, the following sections are numbered from number 1 to 5.

Author response: Thank you for pointing this out. I have renumbered the paragraphs of this paper. The first part of the introduction is numbered 1, and the following parts are numbered 2-6 in order.

  • The second paragraph of the Introduction section is too long to be understood, and I would suggest that divide it into two paragraphs: one is for the research status, another is for virtue of the machine learning method.

Author response: Thank you for pointing this out. I am deeply sorry for the bad reading experience I have caused you. I have divided the second paragraph of the introduction into two paragraphs on page 2: one describing the researcher's exploration of machine learning methods and their current shortcomings. One paragraph introduces the details of this paper's model, how to overcome the shortcomings of the existing methods and the advantages of this paper's model.

  • In the last paragraph of the Introduction, I would suggest to first list the contributions rather than mentioning within the text and separate out the organization at the end.

Author response: Thank you for pointing this out. I have removed the presentation of the contributions in the text and highlighted the main contribution in the introduction and conclusion.

The revised text reads as follows on [line 81 in page 2]: To address the aforementioned issues, this study provides a new discriminative method (PCA-GA-ET) that removes information overlap across data, has a fast-training time, and is highly accurate.

The revised text reads as follows on [line 405 in page 13]: The main contribution of this study is in determining groundwater sources in hydrogeological complex mines using the PCA-GA-ET technique, overcoming the shortcomings of ET in the water source identification problem, and providing a new way of thinking for mine water identification methods using machine learning multi-method fusion.

  • Authors have introduced the potential limitations of existing methods, however, they do not illustrated that how AI can come-in to facilitate those limitations.

Author response: Thank you for pointing this out. Based on the limitations of existing methods mentioned in the introduction, we make use of machine learning methods to complement these deficiencies. We take full advantage of the strengths of these artificial intelligence methods to construct models. The details of model construction I have added in the introduction.

The revised text reads as follows on [line 83 in page 2]: Raw data frequently has flaws that can impair the performance of machine learning models. Redundant features might waste computing resources and impair the model's generalization capabilities. The principal component analysis (PCA) approach is a popular dimensionality reduction technique that combines strongly linked data into fewer new features while eliminating information overlap across features. In terms of model selection, the tree model can handle high-dimensional data and has great robustness, making it resistant to outliers and noise. Because the commonly used random forest technique will cause 20% of the data to enter out-of-bag estimation during the training process, and the small training sample will result in insufficient model learning, we select the extreme tree (ET) algorithm as the discriminant model. Extreme tree is a variant of random forest. It selects features at random for segmentation, which reduces the danger of overfitting. Furthermore, because of the uncertainty of feature selection, extreme trees can be taught faster than random forests. However, the performance of extreme trees is affected by factors such as the depth of the tree and the number of decision trees. The typical manual tuning strategy may overlook some parameter variations, causing the model to perform poorly. The shortcomings of extreme trees are overcome by applying genetic algorithms (GA) to determine the optimal solution of extreme tree parameters in the search space. Genetic algorithms do not require information such as derivatives of the solution function and are thus appropriate for complex nonlinear situations.

In addition I add three advantages of the methodology of this study to the third paragraph of the introduction.

The revised text reads as follows on [line 102 in page 2]: In this study, the PCA-GA-ET model provides the following advantages: (1) quick training time and high recognition efficiency; (2) it can fit the data better when there is less data; (3) PCA-GA-ET can effectively identify water sources and address the problem of complex hydrogeological circumstances.

  • Methods such as PCA, GA and ET has been used to classifying water sources. What is the reason for using the three methods? How these methods are coupled in the study?

Author response: Thank you for pointing this out. Although these strategies have been used to identify water sources, they have only been utilized in isolation. No one has used a combination of those methods to water source discrimination. In this study, the three aspects of data mining, model selection, and reinforcement learning, are employed to build stable and accurate quick discriminative models.

After we obtained these datasets, we first considered whether there were any problems with these data, including the removal of outliers, missing values, and information redundancy. After we calculated the correlation coefficients between the features, we found that two of them had significant information overlap. Therefore, we need to eliminate the information redundancy of the data to ensure that there are no potential problems in the data that would affect the performance of the model. PCA is a popular algorithm that has unique advantages in dimensionality reduction. The choice of the discriminant algorithm is determined by the size of the data. Previously, we tried many machine learning classification algorithms, such as support vector machines, BP artificial neural networks, etc. However, we did not achieve good results due to the size of our data. The tree model can better fit the data of small to medium size. The extreme tree, as an improved algorithm of random forest, has the feature of randomly selecting features to fit the learning samples better. The performance of machine learning algorithms depends on the choice of parameters. The tree depth and the number of classifiers of the extreme tree directly affect the accuracy of the model. We used manual parameter tuning in our previous random forest article, however, we did not achieve good results. In this study, we use automatic parametric tuning. At first, we first considered the particle swarm algorithm (PSO), however, the solution obtained by PSO is floating point type, while the tree depth and the number of classifiers are integer type. PSO is more suitable for optimizing the penalty factor C and the weight g of SVM. genetic algorithm has the advantages of fast search, wide applicability, avoiding getting into local optimal solutions, dealing with high dimensional problems and easy parallelization. Genetic algorithms enable discriminative models to reduce training time and obtain higher accuracy.

After PCA processing, the data are fed into the ET model for training. At this time, the GA finds the optimal values of tree depth and number of classifiers based on the magnitude of the fitness and feeds them to the ET model.

  • Regarding the datasets, it would be nice to see the correlation map between the input features and target classes.

Author response: Thank you for pointing this out. Yes, we can select two variables with similar concentration distributions based on the distribution range of the input features. The relationship density plots of these two variables, such as Ca2+ and Cl-, and the target values are shown in Figure 1. There are three other such relationship density plots, Mg2+ vs. Cl-, Na+ vs. SO42- and Na++K+ vs. HCO3-, respectively.

Figure 1 Bivariate relationship between Ca2+ and Cl- and marginal density (A: surface water B: Quaternary water C: Permian water D: Carboniferous water E: Cambrian water)

  • The format of the manuscript should be improved carefully.

Author response: Thank you for pointing this out. We modified the format of the references to conform to the requirements of the journal. In addition, we enlarged the piper trilinear diagram (Fig 3, Page 7) and the geological pipeline diagram (Fig 2, Page 6). Finally, we revised the syntax of the full text and described Eqs. 8-12 in detail [Line 259 in page 9].

  • There is too much content in the last paragraph, and it should be written concisely.

Author response: Thank you for pointing this out. We have removed unnecessary elements from the conclusions and added some key findings, contributions, and perspectives for future work.

  • What are the limitations of the presented research? And what the implications of the proposed approach? Authors may think of providing a practical use-case demonstrating how this trained model may be used in practice.

Author response: Thank you for pointing this out. The limitations of the proposed method are as follows: First, this technology has not been tried in mines with varying geological conditions. Furthermore, if the chemical concentration of the groundwater is not present in the concentration distribution of the study sample, the approach is invalid. 

The implications of this approach are as follows:The main contribution of this study is in deter-mining groundwater sources in hydrogeological complex mines using the PCA-GA-ET technique, overcoming the shortcomings of ET in the water source identification problem, and providing a new way of thinking for mine water identification methods using machine learning multi-method fusion. it is a worthwhile re-search approach to investigate an accurate, rapid, and stable intelligent system for identifying mine water sources.

The model identified 25 groups of unknown water sources in Pingdingshan coalfield, but only one group was wrong. The accuracy of the model reached 96%.

Reviewer 2 Report

This is the review manuscript “Classification of water inrush source in coal mine based on PCA-GA-ET”. Although the topic is interesting, I could not find any novelty in this paper. As a result, I strongly recommend rejecting the paper directly. 

The paper should be read by an English native speaker.

Author Response

Response to Reviewer 2 Comments

Re: Manuscript ID:water-2351772 and Title:Classification of water source in coal mine based on PCA-GA-ET

Response to Reviewers

Dear reviewer 2:

Thank you for your letter and the reviewer’s comments concerning our manuscript entitled “water-2351772”. Those comments are valuable and very helpful. We have read through comments carefully and have made corrections. Based on the instructions provided in your letter, we uploaded the file of the revised manuscript. Revisions in the text are shown using red highlight for additions, and strikethrough font for deletions. Those changes are highlighted within the manuscript. Please see below, in blue, for a point-by-point response to the reviewer’s comments and concerns. All page numbers refer to the revised manuscript file with tracked changes.

Reviewer's Comments to the Authors:

This is the review manuscript “Classification of water inrush source in coal mine based on PCA-GA-ET”. Although the topic is interesting, I could not find any novelty in this paper. As a result, I strongly recommend rejecting the paper directly.

Author response: Thank you for pointing this out. Thanks again for your busy schedule to review our paper. I think our paper has the following innovations: (1) In this study, an intelligent water source identification method based on multi-method fusion is proposed; (2) This method solves the complicated water source discrimination problem in hydrogeology; (3)PCA-GA-ET eliminates the information redundancy existing in the original data, and we prove that the redundant information affects the model accuracy; (4) We used this method to discriminate 25 groups of unknown water source samples, and only one group was wrong. The accuracy of the model is 96%, so we think this method is feasible in water source discrimination. (5) By calculating Gini coefficient, we can see that Ca2+ ions have strong discriminating ability, so more attention should be paid to the extraction of Ca ions in future research. Limitations of this research method: This method has not been used in different mining areas with different geological conditions, so it can only be used as a reference for mining areas with different geological conditions.

Reviewer 3 Report

The manuscript deals with the important issue of the classification of water inrush source in coal mine based on PCA-GA-ET, as water inrush has hampered regular coal mine mining, and proper identification of the source of water inrush is critical for water damage prevention and control in mines.

Water samples were obtained from 124 groups of important aquifers in order to investigate the source of water inrush in the Pingdingshan coalfield. The main water-filled aquifers in Pingdingshan coalfield include surface water (I), Quaternary pore water (II), Carboniferous limestone karst water (III), Permian sandstone water (IV) and Cambrian limestone karst water (IV). Authors as to avoid the impact of  information overlap on the accuracy of the model used principal component analysis to extract the main indicators. Authors constructed water source identification model based on the adaptive differential evolution genetic algorithm to optimize the super parameters of the extreme tree. Authors verify the reliability of the model, PCA-GA-ET was compared with grid search-random forest, artificial neural network, and particle swarm optimization-support vector machine. The use of principal component analysis can reduce the information overlap of data, and the extreme tree model optimized by genetic algorithm greatly improves the efficiency and accuracy of water inrush source identification. Comments and suggestions: In the abstract in the following sentence something lacks: model. Principal. Why did you choose these particular methods for the analysis? It is important to present the general framework of the study in the introduction. Could you please explain why this approach is the best solution for performing the analysis? The contribution of your research should be further emphasized with more details. All the equations should be given more detailed way. In the introduction the choice of the reference should be supplemented with respect to the categorization is based on the distinctive attributes of each factor, the factors within a particular layer are interdependent, in that they either depend on the factors in the higher layer or have an impact on these higher-layer factors, while also exerting dominance over the factors in the lower layer or being subject to the influence of the lower-layer factors, as in eg. Most Searched Topics in the Scientific Literature on Failures in Photovoltaic Installations. Energies 2022, 15, 8108.The figure presenting the piper diagram of hydrochemistry of Pingdingshan coal water sample should be bigger. The caption of the geological structure map of Pingdingshan coalfield is too small. Lack of numbering the lines in the whole manuscript. In fourth section (4. PCA-GA-ET model performance verification) only one subsection is included, as 4.1. Variable significance analysis, so the title of this subsection can be omitted. Lack of discussion about possible limitations of using the proposed approach. What is the added value  and novelty of the paper? Please add some perspectives of future work. 

Sth lacks in this sentence:

. To avoid the impact of information overlap on the accuracy of the model. 

Author Response

Response to Reviewer 3 Comments

Re: Manuscript ID:water-2351772 and Title:Classification of water source in coal mine based on PCA-GA-ET

Response to Reviewers

Dear reviewer 3:

Thank you for your letter and the reviewer’s comments concerning our manuscript entitled “water-2351772”. Those comments are valuable and very helpful. We have read through comments carefully and have made corrections. Based on the instructions provided in your letter, we uploaded the file of the revised manuscript. Revisions in the text are shown using red highlight for additions, and strikethrough font for deletions. Those changes are highlighted within the manuscript. Please see below, in blue, for a point-by-point response to the reviewer’s comments and concerns. All page numbers refer to the revised manuscript file with tracked changes.

Reviewer's Comments to the Authors:

Reviewer 3

The manuscript deals with the important issue of the classification of water inrush source in coal mine based on PCA-GA-ET, as water inrush has hampered regular coal mine mining, and proper identification of the source of water inrush is critical for water damage prevention and control in mines.

Water samples were obtained from 124 groups of important aquifers in order to investigate the source of water inrush in the Pingdingshan coalfield. The main water-filled aquifers in Pingdingshan coalfield include surface water (I), Quaternary pore water (II), Carboniferous limestone karst water (III), Permian sandstone water (IV) and Cambrian limestone karst water (IV). Authors as to avoid the impact of information overlap on the accuracy of the model used principal component analysis to extract the main indicators. Authors constructed water source identification model based on the adaptive differential evolution genetic algorithm to optimize the super parameters of the extreme tree. Authors verify the reliability of the model, PCA-GA-ET was compared with grid search-random forest, artificial neural network, and particle swarm optimization-support vector machine. The use of principal component analysis can reduce the information overlap of data, and the extreme tree model optimized by genetic algorithm greatly improves the efficiency and accuracy of water inrush source identification. Comments and suggestions:

  • In the abstract in the following sentence something lacks: model. Principal.

Author response: Thank you for pointing this out. I have improved the sentence that the reviewer pointed out in the abstract.

The revised text reads as follows on [line 21 in page 1]: The studies reveal that PCA-GA-ET can eliminate information overlap between data and simplify the data structure, therefore improving the efficiency and accuracy of water source detection. We discovered that by utilizing the evolutionary algorithm to optimize parameters like the depth of the extreme trees and the number of decision trees, we could get the model to converge faster, be more stable, and be more accurate.

  • Why did you choose these particular methods for the analysis?

Author response: Thank you for pointing this out. After we obtained these datasets, we first considered whether there were any problems with these data, including the removal of outliers, missing values, and information redundancy. After we calculated the correlation coefficients between the features, we found that two of them had significant information overlap. Therefore, we need to eliminate the information redundancy of the data to ensure that there are no potential problems in the data that would affect the performance of the model. PCA is a popular algorithm that has unique advantages in dimensionality reduction. The choice of the discriminant algorithm is determined by the size of the data. Previously, we tried many machine learning classification algorithms, such as support vector machines, BP artificial neural networks, etc. However, we did not achieve good results due to the size of our data. The tree model can better fit the data of small to medium size. The extreme tree, as an improved algorithm of random forest, has the feature of randomly selecting features to fit the learning samples better. The performance of machine learning algorithms depends on the choice of parameters. The tree depth and the number of classifiers of the extreme tree directly affect the accuracy of the model. We used manual parameter tuning in our previous random forest article, however, we did not achieve good results. In this study, we use automatic parametric tuning. At first, we first considered the particle swarm algorithm (PSO), however, the solution obtained by PSO is floating point type, while the tree depth and the number of classifiers are integer type. PSO is more suitable for optimizing the penalty factor C and the weight g of SVM. genetic algorithm has the advantages of fast search, wide applicability, avoiding getting into local optimal solutions, dealing with high dimensional problems and easy parallelization. Genetic algorithms enable discriminative models to reduce training time and obtain higher accuracy.

  • It is important to present the general framework of the study in the introduction.

Author response: Thank you for pointing this out. I have added the overall framework of the model in the third paragraph of the introduction.

The revised text reads as follows on [line 83 in page 2]: Raw data frequently has flaws that can impair the performance of machine learning models. Redundant features might waste computing resources and impair the model's generalization capabilities. The principal component analysis (PCA) approach is a popular dimensionality reduction technique that combines strongly linked data into fewer new features while eliminating information overlap across features. In terms of model selection, the tree model can handle high-dimensional data and has great robustness, making it resistant to outliers and noise. Because the commonly used random forest technique will cause 20% of the data to enter out-of-bag estimation during the training process, and the small training sample will result in insufficient model learning, we select the extreme tree (ET) algorithm as the discriminant model. Extreme tree is a variant of random forest. It selects features at random for segmentation, which reduces the danger of overfitting. Furthermore, because of the uncertainty of feature selection, extreme trees can be taught faster than random forests. However, the performance of extreme trees is affected by factors such as the depth of the tree and the number of decision trees. The typical manual tuning strategy may overlook some parameter variations, causing the model to perform poorly. The shortcomings of extreme trees are overcome by applying genetic algorithms (GA) to determine the optimal solution of extreme tree parameters in the search space. Genetic algorithms do not require information such as derivatives of the solution function and are thus appropriate for complex nonlinear situations.

  • Could you please explain why this approach is the best solution for performing the analysis?

Author response: Thank you for pointing this out. I think there are three defects in the current research: (1) When the amount of data is large, the calculation amount of mathematical model will increase, so the discrimination speed is slow, such as distance discrimination method and Fisher discrimination method; (2) When the hydrogeology is complicated, the hydrochemical method is not effective in identifying water sources, such as Piper diagram; (3) The hidden problem of the original data has not been solved, which affects the performance of the model. The model designed in this paper can make up for these defects. In addition, we have calculated more classification indexes of different methods, as shown in Table 1, which clearly shows that PCA-GA-ET is much better than MLP, GA and RF.

Tabel 1 The classification indexes of different methods when training

Model

Macro

Weight

Micro

Kappa

F1-score

AUC

PCA-GA-ET

0.975

0.965

0.960

0.946

0.919

0.977

RF

0.969

0.932

0.920

0.883

0.921

0.965

SVM

0.513

0.650

0.640

0.502

0.511

0.788

MLP

0.750

0.770

0.720

0.643

0.697

0.806

  • The contribution of your research should be further emphasized with more details.

Author response: Thank you for pointing this out. We have added more details about the contribution in the conclusion.

The revised text reads as follows on [line 405 in page 13]: The main contribution of this study is in determining groundwater sources in hydrogeological complex mines using the PCA-GA-ET technique, overcoming the shortcomings of ET in the water source identification problem, and providing a new way of thinking for mine water identification methods using machine learning multi-method fusion.

  • All the equations should be given more detailed way.

Author response: Thank you for pointing this out. We added more explanation to the coefficient matrix of the principal component analysis.

The revised text reads as follows on [line 121 in page 3]: Among them, ,,... represents the new main component, ,,... represents the original characteristics, and ,,... represents linear combination coefficient.

The revised text reads as follows on [line 151 in page 4]: Among then,  is coding accuracy. Let a parameter take a range of values , an interval length -, and a string length .

The revised text reads as follows on [line 154 in page 4]: Then, x is the optimal solution obtained, Umin is the minimum solution that can be obtained by the parameter, and bi is the binary number of the optimal solution.

The revised text reads as follows on [line 258 in page 8]: Where,  is the standardized value of variables,  is the sample value before standardization,  is the maximum value of sample data, and  is the minimum value of sample data.

The revised text reads as follows on [line 274 in page 9]: where Y1, Y2, Y3, Y4, Y5 are freshly created features and X1, X2, X3, X4, X5, X6 are Na++K+, Ca2+, Mg2+, Cl-, SO42-, and HCO3-.

The revised text reads as follows on [line 310 in page 10]: Where, RMSE is root mean square error, N is the number of observations,  is the predicted value, and  is the true value.

  • In the introduction the choice of the reference should be supplemented with respect to the categorization is based on the distinctive attributes of each factor, the factors within a particular layer are interdependent, in that they either depend on the factors in the higher layer or have an impact on these higher-layer factors, while also exerting dominance over the factors in the lower layer or being subject to the influence of the lower-layer factors, as in eg. Most Searched Topics in the Scientific Literature on Failures in Photovoltaic Installations. Energies 2022, 15, 8108.

Author response: Thank you for pointing this out. We have cited some literature on classification.

The revised text reads as follows on [line 51 in page 2]: Water source discrimination can be seen as a classification problem (Kut P 2022; Nishitsuji Y 2019; Feng R 2020; Caté A 2018).

  • The figure presenting the piper diagram of hydrochemistry of Pingdingshan coal water sample should be bigger. 

Author response: I have made the piper diagram larger in page 7.

  • The caption of the geological structure map of Pingdingshan coalfield is too small. Lack of numbering the lines in the whole manuscript.

Author response: Thank you for pointing this out. I made the geological structure map of Pingdingshan coalfield bigger in page 6. Meantime, I add line numbers to the full text.

  • In fourth section (4. PCA-GA-ET model performance verification) only one subsection is included, as 4.1. Variable significance analysis, so the title of this subsection can be omitted. 

Author response: I deleted the title of 4.1.

  • Lack of discussion about possible limitations of using the proposed approach.

Author response: Thank you for pointing this out. I have added in the conclusion about the limitations of this method.

The revised text reads as follows on [line 409 in page 13]: Despite the fact that the source of mine water has been established, there are still some issues. Firstly, this technology has not been tried in mines with varying geological conditions. Secondly, if the chemical concentration of the groundwater is not present in the concentration distribution of the study sample, the approach is invalid.

  • What is the added value and novelty of the paper? Please add some perspectives of future work.

Author response: Thank you for pointing this out. This study's value and innovation can be classified into four categories: (1). An intelligent water source discrimination model is proposed; (2) Multiple machine learning algorithms are used to differentiate water source kinds, and the water source types are detected more quickly and accurately; (3) This model solves the problem of locating groundwater in complex hydrogeological circumstances; (4) With a small amount of data, this approaches attain high accuracy.

We have added to the conclusion a view of the future work.

The revised text reads as follows on [line 413 in page 13]: The following research should attempt to apply machine learning methods to different mining sites with different geological conditions, as well as use GIS methods to understand the geological conditions surrounding the water samples and closely integrate the hydrological conditions of groundwater. Furthermore, the sample database should be built up by collecting as many samples as feasible. The model learns enough from the sample database to increase generalization and, to a large part, avoid mine water breakout mishaps. Finally, it is a worthwhile research approach to investigate an accurate, rapid, and stable intelligent system for identifying mine water sources.

Round 2

Reviewer 2 Report

Please consider the decision letter.

Reviewer 3 Report

Accept in present form.